# Coronary Physiology in the Cardiac Catheterization Laboratory

**DOI:** 10.3390/jcm8020255

**Published:** 2019-02-18

**Authors:** Samit M. Shah, Steven E. Pfau

**Affiliations:** 1Yale-New Haven Hospital, Yale School of Medicine, New Haven, CT 06510, USA; steven.pfau@yale.edu; 2VA Connecticut Healthcare System, West Haven, CT 06516, USA

**Keywords:** coronary artery disease, cardiovascular physiology, microvascular angina

## Abstract

Coronary angiography has been the principle modality for assessing the severity of atherosclerotic coronary artery disease for several decades. However, there is a complex relationship between angiographic coronary stenosis and the presence or absence of myocardial ischemia. Recent technological advances now allow for the assessment of coronary physiology in the catheterization laboratory at the time of diagnostic coronary angiography. Early studies focused on coronary flow reserve (CFR) but more recent work has demonstrated the physiologic accuracy and prognostic value of the fractional flow reserve (FFR) and instantaneous wave free ratio (iFR) for the assessment of coronary artery disease. These measurements have been validated in large multi-center clinical trials and have become indispensable tools for guiding revascularization in the cardiac catheterization laboratory. The physiological assessment of chest pain in the absence of epicardial coronary artery disease involves coronary thermodilution to obtain the index of microcirculatory resistance (IMR) or Doppler velocity measurement to determine the coronary flow velocity reserve (CFVR). Physiology-based coronary artery assessment brings “personalized medicine” to the catheterization laboratory and allows cardiologists and referring providers to make decisions based on objective findings and evidence-based treatment algorithms. The purpose of this review is to describe the theory, technical aspects, and relevant clinical trials related to coronary physiology assessment for an intended audience of general medical practitioners.

## 1. Introduction

The first coronary angiogram nearly 60 years ago provided a simplified glimpse of a dynamic and highly regulated process—the delivery of blood to working myocardium. Sones’ selective coronary angiography launched a new field of diagnosis and treatment for atherosclerotic coronary obstruction [1]. Since that time our understanding of the physiology underlying myocardial ischemia has increased rapidly [2], and while invasive coronary angiography remains the “gold standard” its limitations are well-documented [3,4]. There is a complex relationship between the anatomic severity of luminal narrowing and the presence of myocardial ischemia or risk of subsequent myocardial infarction [5,6] and angiography alone is unable to determine whether a luminal stenosis causes ischemia or anginal symptoms. Technological advances in the last 30 years now allow direct, real time measurement of coronary flow and pressure in individual patients and this data can be applied directly at the time of coronary angiography to guide treatment of individual atherosclerotic lesions. Most recently, real time assessment of coronary physiology is being applied to conditions that affect coronary blood flow outside the realm of atherosclerotic luminal obstruction, providing insight into a variety of disease states such as endothelial dysfunction and microvascular disease. The purpose of this review is to describe the theory, technical aspects, and relevant clinical trials related to coronary physiology assessment for an intended audience of general medical practitioners.

## 2. Historical Basis for Assessing Coronary Physiology in the Catheterization Laboratory

With exertion, coronary blood flow can increase to 300–400% of resting levels [7], both in animal models and in humans during treadmill exercise [8]. In 1974 Gould et al. described the concept of coronary flow reserve (CFR), a ratio of maximal flow to resting flow in response to increases in myocardial oxygen demand [5]. Using an in vitro animal model, the authors demonstrated that resting coronary blood flow is not significantly affected by a stenosis until the lumen is at least 85% occluded. However, during hyperemia (achieved with a vasodilatory contrast dye bolus) the maximal achievable coronary blood flow was limited by a luminal stenosis as low as 30–40% (Figure 1). This seminal study demonstrated that the significance of a coronary stenosis is directly related to coronary blood flow, and that the magnitude of the reduction in blood flow is most clinically relevant during maximal coronary blood flow (or hyperemia). Questions regarding the clinical correlations and prognostic implications of the physiological assessment of an anatomic coronary stenosis set the stage for decades of further investigation.

The first coronary angioplasty on an awake patient was performed on 14th September 1977 by Andreas Gruentzig in Zurich, Switzerland [10,11]. The operators used a double lumen catheter that allowed for intra-coronary pressure measurement as well as angioplasty balloon dilatation. In their initial report Gruentzig and his colleagues documented the angiographic findings as well as the hemodynamic significance of the stenosis as represented by the arterial pressure proximal and distal to the stenosis (Figure 2). After balloon angioplasty the pressure gradient was reduced from 58 mmHg to 19 mmHg, which correlated with a significant decrease in the severity of the angiographic stenosis. The authors noted that an “increase in distal coronary pressure” after balloon angioplasty could be utilized to gauge the success of the procedure. The concept of improving blood flow to the myocardium by relieving a hemodynamically significant stenosis was intuitive. However, the Gruentzig balloon dilatation catheter had a large outer diameter and contributed to the luminal obstruction of a stenosis, confounding the assessment of distal coronary pressure. As a result of the physical diameter of the pressure measurement catheter, the “translesional pressure gradient” that was measured at the time of angioplasty did not accurately correlate with the severity of stenosis measured angiographically [12]. Anatomic evaluation with qualitative and quantitative angiography remained the cornerstone of the assessment of coronary artery stenosis severity, guiding decisions regarding revascularization with angioplasty or coronary artery bypass graft surgery, and evaluating the outcome of revascularization interventions [13]. Still, as noted by Gruentzig’s team, the idea that “anatomy alone” did not “predict the physiologic consequence of individual stenoses” persisted [12]. 

The measurement of human coronary blood flow and CFR in conscious humans became possible with the advent of catheter-based pulsed Doppler technology [14,15,16]. The underlying principle of Doppler-based measurement is that the frequency shift that occurs when transmitted ultrasound waves are reflected off of moving red blood cells is proportional to the velocity of blood flow. This measured frequency shift is converted to a calculated flow velocity which approximates the volumetric blood flow velocity [17]. Intra-coronary Doppler flow velocity measurement accurately correlated with absolute flow velocity in in vitro systems as well as with Doppler velocity measured with extravascular epicardial probes [17]. However, due to concern that even ~1 mm microcatheters might contribute coronary luminal obstruction and prohibit accurate flow assessment in patients with coronary stenosis, the 0.018 inch (0.45 mm) diameter coronary Doppler guidewire was developed [18]. The “Doppler wire” could measure coronary flow velocity distal to a stenosis at rest and after the induction of hyperemia using papaverine, adenosine, dipyramidole, or contrast media. In spite of the Doppler wire being validated for the accurate assessment of CFR there were significant caveats to Doppler wire-derived physiological assessment of coronary stenosis. CFR measurement encompasses the entire coronary circulation, including the epicardial coronary artery and the microvascular bed, thus the specificity for determining the contribution of an epicardial stenosis to a reduction in Doppler flow velocity may be decreased in patients with microvascular disease. Additionally, resting Doppler flow velocity measurements are susceptible to hemodynamic factors that may alter coronary flow velocity, such as changes in heart rate, blood pressure, and left ventricular function, which in turn reduce the accuracy of CFR measurement. Finally, Doppler wire-derived CFR guided percutaneous coronary intervention has been shown to have no difference in outcomes compared to angiography-guided intervention [19]. These limitations, combined with technical challenges with operating the Doppler wire, limited adoption of this technology for routine physiological assessment in the catheterization laboratory.

Nearly 15 years after Gruentzig’s initial report of the translesional pressure gradient, Nico Pijls and Bernard DeBruyne in the Netherlands investigated the application of an 0.015 inch diameter (0.38 mm) coronary guidewire that was capable of measuring a pressure gradient across a stenosis. Using this “pressure wire” they were able to experimentally validate the relationship between coronary pressure and coronary flow, both at rest and during maximal hyperemia [21]. This seminal study laid the foundation for the measurement of flow reserve by pressure rather than Doppler velocity. There was a strong correlation between coronary blood flow by Doppler wire assessment and the ratio of the distal coronary pressure (P_d_) to the proximal aortic pressure (P_a_). During hyperemia, when microcirculatory resistance is negligible, the P_d_ represents the maximal coronary flow in the distal vessel and the P_a_ represents the maximal coronary flow if the vessel were normal. The term “fractional flow reserve” (FFR) was introduced to describe the ratio of P_d_ to P_a_, or the maximal achievable flow in the presence of a stenosis divided by the maximum expected flow if the stenosis were absent. In other words, the FFR represents the hemodynamic contribution of a coronary stenosis to the reduction of blood flow in a coronary artery territory. While this was reminiscent of Gruentzig’s translesional pressure gradient, Pijls and De Bruyne advanced the field of coronary physiology by using a small diameter coronary “pressure wire”, demonstrating the physiological importance of minimizing microvascular resistance with pharmacologic hyperemia, and detailing the relationship of venous filling pressure and the collateral circulation to coronary blood flow. 

These animal validations were followed by clinical studies comparing FFR measurement to non-invasive measures of ischemia including exercise stress testing, thallium scintigraphy, and dobutamine stress echocardiography [22]. Patients who presented with angina underwent non-invasive stress testing and subsequently coronary angiography with pressure wire assessment. An FFR of 0.75, or a distal coronary pressure of less than 75% of the expected pressure, indicated a coronary stenosis that was “significant”, or in other words every patient with an FFR less than 0.75 was found to have evidence of ischemia by a non-invasive test. After revascularization the non-invasive tests were repeated and showed no evidence of ischemia (Figure 3). Furthermore, patients with an FFR greater than 0.75 were managed with medical therapy and suffered no events with a mean follow-up of 14.5 months. This landmark study demonstrated that measurement of the FFR reliably defines coronary lesions that cause myocardial ischemia independent of the angiographic severity of stenosis.

## 3. Coronary Physiology in the Evaluation of Coronary Artery Disease

The validation of FFR as an invasive determinant of ischemia led to a series of important randomized trials that have altered the practice of interventional cardiology. The multi-center DEFER trial investigated whether patients with a non-ischemic FFR of greater than 0.75 could safely defer percutaneous coronary intervention (PCI) regardless of the angiographic severity of a stenosis [23]. The study design included 325 patients who were referred for coronary angiography and were planned to undergo PCI. FFR was performed on all patients; if the FFR was greater than 0.75, patients were randomized to deferral of intervention or performance of PCI as planned. If the FFR was less than 0.75 patients underwent PCI as planned. For patients with an FFR greater than 0.75 there was no difference in event-free survival at 24 months if PCI was performed or deferred, and patients who were randomized to deferring PCI reported less angina. At 5 years there was no difference in outcome if PCI was deferred [24] for an FFR greater than 0.75. This established the safety of deferring coronary intervention in coronary arteries with an FFR above the physiological threshold for ischemia. 

The international multi-center FAME trial compared FFR-guided coronary revascularization to angiographic selection in patients with multi-vessel coronary artery disease and a stenosis of at least 50% [25]. Patients who were randomized to the FFR-guided arm underwent intervention only if the FFR was 0.80 or less, and patients in the angiography-guided arm underwent intervention based on existing standard of care of angiographic severity. This trial enrolled 1005 patients across 20 centers in the United States and Europe and a total of 2415 stents were placed. At one-year patients in the FFR arm were less likely to suffer death, MI, and repeat revascularization; and the overall rate of death or MI at one year was 11.1% in the angiography-guided arm compared to 7.3% in the FFR group. A sub-analysis of the FFR-guided revascularization arm reported the FFR values for tertiles of angiographic stenosis (Figure 4). In lesions of 50–70% stenosis, 35% had an ischemic FFR; in 71–90% stenoses, 80% had an ischemic FFR; and in 91–99% stenoses, 96% had an ischemic FFR [26]. This showed that in moderate severity stenoses there was significant heterogeneity with regard to the presence of ischemia, and even in angiographically severe stenoses (>70%) up to 20% of patients did not have physiologic evidence of ischemia [14]. Overall, the FAME trial showed that using FFR to guide PCI reduced adverse cardiovascular events by approximately 30% and the estimated number needed to treat to prevent one adverse event was 20 patients. 

FAME II compared FFR-guided PCI to the best available medical therapy in patients with stable coronary artery disease [27]. Patients with FFR less than 0.80 were randomized to PCI or medical therapy alone. The trial was stopped early due to a significantly higher incidence of urgent revascularization in the medical therapy arm. At three years of follow-up the rate of urgent revascularization and symptomatic angina remained significantly lower in the FFR-guided PCI group compared with medical therapy [28]. Furthermore, there was no significant difference in cost between the FFR-guided PCI and medical therapy groups. Thus, the FAME studies demonstrated the superiority of FFR-guided PCI as compared with angiographic-guided intervention or medical therapy.

FFR has been also validated for the physiological assessment of non-culprit stenoses in patients who present with acute coronary syndromes [29]. Theoretically, microvascular obstruction from ischemia or infarction may alter the hyperemic response to adenosine but FFR remains a valid assay. Multiple studies have validated FFR for assessment of non-culprit stenoses in the setting of acute coronary syndromes or after myocardial infarction [30]. However, there is evidence that using FFR to defer intervention to a culprit stenosis may be associated with a worse outcome due to increased major adverse cardiovascular events in follow-up [31].

The ORBITA trial, first published in 2017, was a blinded, multi-center study across 5 sites in the United Kingdom that compared PCI to sham intervention in patients with stable angina and >70% angiographic stenosis [32]. All patients were treated with optimal medical therapy. The primary endpoint was treadmill exercise time at six weeks; the study found no significant difference between the PCI and sham intervention groups and concluded that there was no benefit to PCI in stable angina. However as a secondary assessment all patients underwent physiological testing with FFR and the instantaneous wave-free ratio (iFR) at the time of coronary angiography [33]. The mean FFR was 0.69 ± 0.16 and the mean iFR was 0.76 ± 0.22. Patients who were randomized to PCI were found to have a significant improvement in post-intervention dobutamine stress echocardiography wall motion compared with patients who did not undergo intervention, and the magnitude of the effect correlated with the severity of ischemia by FFR and iFR. However, pre-intervention FFR or iFR did not predict whether patients experienced improvement in angina after intervention. The ORBITA trial and subsequent secondary analysis stratified by FFR demonstrated the efficacy of PCI for relieving objectively documented ischemia in physiologically significant coronary stenoses [33].

Despite the robust evidence and guideline recommendation for the routine use of FFR for assessing intermediate-severity stenoses, the overall rate of FFR use remains low [34,35]. In part, this may be due to the technical limitations of FFR—namely, a slightly increased procedure time, obligation to use a pressure-sensor tipped guidewire, and the financial expense of using adenosine to induce hyperemia [14]. Furthermore, many patients find intravenous adenosine uncomfortable and report symptoms of chest pain or dyspnea during the infusion, which may last up to four minutes. While intracoronary injection of adenosine is routinely utilized therapeutically during acute coronary syndromes or coronary intervention to reduce microcirculatory dysfunction, it is less well tolerated in stable patients during physiological assessment [36,37]. To overcome these limitations novel indices of resting pressure have been developed including the iFR. iFR is measured with a pressure-sensor tipped guidewire (Royal Philips, Amsterdam) and using a proprietary algorithm measures the proximal and distal coronary pressure during the phase of diastole when microcirculatory resistance is at a theoretical minimum (the “wave free” period) [38]. Since the pressure measurement is obtained during a period of minimal microcirculatory resistance inducing maximal hyperemia with adenosine is not necessary. In many situations iFR closely approximates FFR and has concordant results with non-invasive stress testing [39,40], and an iFR cutoff value of 0.89 has an accuracy of 80% for identifying lesions with an FFR less than 0.80 [40]. However, simulations and validation studies in humans show that iFR may correlate best with FFR for physiologically insignificant (or “FFR negative”) lesions [41]. In 2017 two large multi-center clinical trials were published comparing iFR and FFR. The DEFINE-FLAIR study was an industry-sponsored (Royal Philips) randomized trial comparing iFR-guided or FFR-guided coronary revascularization [42]. At 1-year iFR-guided revascularization was found to be non-inferior to FFR and patients in the iFR reported fewer symptoms in the procedural period (due to the omission of adenosine). Furthermore, procedure time was decreased by nearly 5 min. A second concurrent Philips-sponsored study, “iFR SWEDEHEART,” was performed using the Swedish Coronary Angiography and Angioplasty Registry [43]. A total of 2037 patients with stable angina or an acute coronary syndrome were randomized to undergo iFR or FFR guided revascularization. At 1-year iFR was again shown to be non-inferior to FFR and a higher proportion of patients in the FFR group reported chest pain during the procedure (related to adenosine infusion). A meta-analysis of both trials showed numerically higher myocardial infarctions and deaths in the iFR-guided revascularization group but without statistical significance [44]. A more recent meta-analysis reported iFR to have similar diagnostic performance to FFR [45] and additional studies have also shown that iFR is non-inferior to FFR for the assessment of non-culprit stenoses in acute coronary syndromes [46]. Long-term outcomes studies are ongoing but due to its convenience and promising preliminary data, iFR has become incorporated into routine clinical practice for the physiological assessment of coronary stenoses. 

## 4. Assessment of Microvascular Disease and Endothelial Dysfunction

Chest pain is one of the most common presenting complaints in outpatient visits and to the emergency department [47,48]. Of the patients who are referred for coronary angiography, with or without ischemia on non-invasive stress testing, 20–50% are found to have angiographically normal coronary arteries [3,49,50]. Despite the absence of epicardial coronary artery disease by angiography many patients suffer recurrent presentations for chest pain [48,50]. There is increasing awareness of pathologies beyond obstructive coronary stenosis that can cause myocardial ischemia, including diffuse epicardial coronary artery disease, occlusion of small secondary branches, microvascular disease, or low baseline resting flow [51,52].

The coronary arteries consist of epicardial conduit vessels (>400 µm diameter), microvascular resistance vessels (100–400 µm), and arterioles (<100 µm) [53,54,55]. FFR and iFR are commonly performed as guideline recommended tests for evaluating the hemodynamic significance of epicardial coronary artery disease [14,56,57,58]. However, these measurements do not evaluate the coronary microcirculation and more than 80% of the coronary resistance is determined by the microvascular vessels. One proposed mechanism for chest pain in the absence of epicardial coronary artery disease is abnormal microvascular function [59,60,61,62,63]. Two methods have been developed for the assessment of microvascular resistance: thermodilution index of microcirculatory resistance (IMR) and Doppler wire-derived hyperemic microvascular resistance (HMR). 

Coronary thermodilution is an established method for measuring the CFR, which in the absence of an epicardial stenosis estimates the microvascular resistance [64]. This can be performed with a commercially available coronary pressure wire (Pressure Wire X, Abbott Vascular, Illinois) that has a dual pressure sensor and thermistor at the distal tip. CFR is affected by hemodynamic variables such as heart rate and blood pressure, which affect coronary blood flow. The accuracy of microvascular assessment has been improved by the derivation of the index of microcirculatory resistance (IMR) which incorporates the distal coronary pressure to correct for hemodynamic variability [65,66]. To estimate the flow in a coronary artery 3 mL of room temperature saline is injected into the artery. A thermodilution curve is obtained which demonstrates the change in temperature at the distal temperature sensor over time, and this is used to derive the mean transit time (T_mn_) of blood from the proximal temperature sensor (proximal coronary artery) to the distal sensor (distal coronary artery). Since the volume of blood is static, the T_mn_ is an inverse correlate of coronary blood flow (in other words, the time required for temperature change is longer in vessels with slow flow and shorter in vessels with brisk flow). When measured at rest and during hyperemia, the ratio of the hyperemic flow (1/T_mn_) to the resting flow is the CFR [64]. Using Ohm’s law where resistance is equal to the pressure gradient divided by flow, the IMR can be calculated which represents the coronary microvascular resistance. To calculate IMR pressure is the distal coronary pressure (P_d_) and, as described, flow is estimated as the inverse of the T_mn_. Thus, if resistance is Pd divided by (1/T_mn_), then multiplying the distal coronary pressure (P_d_) by the hyperemic T_mn_ results in the IMR [67]. IMR is obtained during maximal hyperemia with adenosine and has been validated as a reproducible metric of the coronary microcirculatory resistance that is independent of epicardial coronary artery disease [68,69,70]. In the absence of epicardial coronary artery disease ischemic chest pain symptoms have been correlated with an elevated IMR, representing endothelium-independent microvascular dysfunction [49]. In the setting of acute myocardial infarction IMR has been shown to be a more sensitive indicator of microvascular pathology than standard coronary angiography or CFR, and elevated IMR correlates with non-invasive cardiac magnetic resonance imaging (MRI) features of microvascular damage [71,72].

The coronary Doppler-wire derived CFR (or coronary flow velocity reserve, CFVR) reflects epicardial vasomotion and microvascular resistance [73]. This is calculated by recording the average peak Doppler velocity (APV) at hyperemia and at rest. The ratio of the hyperemic velocity to the resting velocity, or CFR, represents the entire coronary circuit (including epicardial and microvascular resistance) [74]. A microvascular-specific measurement, HMR, can be calculated with simultaneous measurement of the distal coronary pressure and the average peak Doppler velocity. The ratio between the mean distal coronary pressure during hyperemia and the hyperemic average peak Doppler velocity is the HMR, which is a derivation of the microvascular resistance using Poiseuille’s Law (resistance is equal to pressure divided by flow). In practice, HMR is calculated as the P_d_ divided by the Doppler velocity. HMR is able to accurately identify patients with microvascular obstruction after myocardial infarction [75]. Furthermore, while IMR must be performed with intravenous adenosine, HMR can be obtained during acetylcholine infusion allowing for simultaneous assessment of endothelial function [76]. 

A comparison study of HMR and IMR showed that the two indices have modest correlation and HMR may be more representative of the overall CFR, but there is no significant difference in the performance of either assay for diagnosing microvascular obstruction after myocardial infarction (as documented by magnetic resonance imaging) [77]. Thermodilution derived flow reserve and IMR were recently compared to Doppler wire derived flow reserve and HMR against the “gold standard” of position emission tomography (PET) derived CFR. While Doppler CFR was found to have better correlation with PET CFR, thermodilution CFR was more reproducible and correlated with PET CFR in cases of abnormally low flow reserve [78].

## 5. Summary and Conclusions

Remarkable conceptual and technological breakthroughs in the understanding of coronary physiology have occurred in the 40 years since Andreas Gruentzig measured a translesional pressure gradient during the first coronary angioplasty. We are now able to assess the physiological and prognostic significance of individual coronary lesions “on the fly” in the catheterization laboratory, and target coronary intervention to those lesions that are most likely to cause future events. In patients who present with acute coronary syndromes and are found to have multi-vessel disease, we can safely assess the physiologic significance of non-culprit coronary stenoses during revascularization of culprit lesions [29,79]. Furthermore, in patients who present for evaluation of chest pain and are found to have an absence of coronary artery disease, we can estimate the resistance (i.e., the physiological significance) of those microcirculatory vessels that cannot be visualized angiographically. 

Physiology-based coronary artery assessment brings “personalized medicine” to the catheterization laboratory and allows cardiologists and referring providers to make decisions based on objective findings and evidence-based treatment algorithms rather than subjective angiographic interpretation. In the near future, computational analysis of the coronary angiogram itself may incorporate a physiological assessment—without guidewires or hyperemic stimuli. These emerging technologies, including invasive quantitative coronary angiography (QFR) and non-invasive CT angiography-based FFR [80], are extrapolations of the foundational investigations in the catheterization laboratory described here. Convenience, accessibility and cost will make non-invasive technologies appealing as the next frontier in the assessment of coronary physiology. 

## Figures and Tables

**Figure 1 jcm-08-00255-f001:**
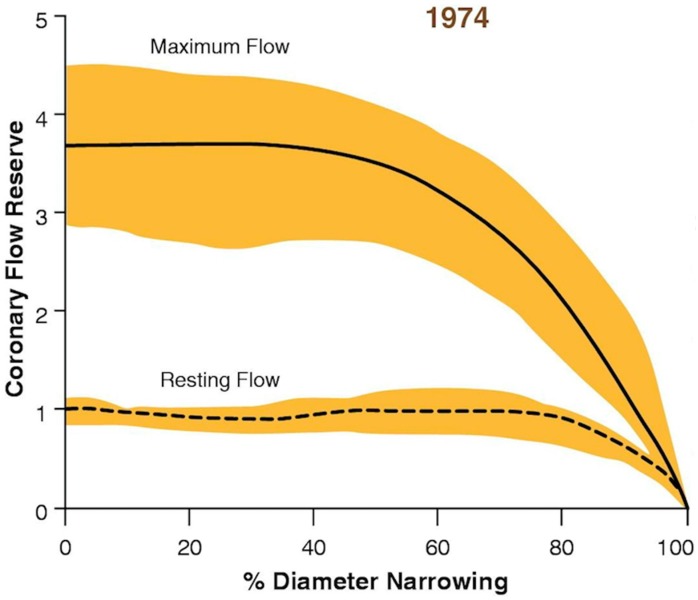
Effect of coronary stenosis on coronary flow reserve. Reprinted with permission from Gould 2009, adapted from Gould et al. 1974 [9]. Experimental model of coronary blood flow at rest and the maximal achievable flow. In the presence of a stenosis resting flow is not limited until a coronary stenosis reaches 85% luminal obstruction. However, maximal achievable flow is limited at stenoses as low as 30–40%. The ratio of the maximal achievable coronary blood flow to the resting blood flow is the coronary flow reserve (CFR).

**Figure 2 jcm-08-00255-f002:**
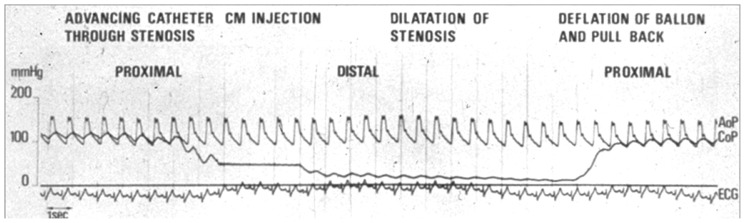
Coronary Artery Translesional Pressure Measurement in the First Angioplasty (1978). Reprinted and adapted from EuroIntervention Vol 13/number 1, Meier, B, “*His master’s art, Andreas Grüntzig’s approach to performing and teaching coronary angioplasty”*, Pages 15–27, Copyright (2017), with permission from Europa Digital & Publishing [20]. Hemodynamic assessment of coronary stenosis during the first coronary angioplasty by Dr. Andreas Gruentzig in 1978. The pressure tracing on top labeled AoP is the aortic pressure. The bottom tracing CoP is the coronary pressure at the distal tip of the pressure-monitoring lumen of the balloon dilatation catheter. Distal to the stenosis there is a significant drop in the coronary pressure that is improved after balloon inflation and dilatation of the stenosis. This was the initial report of the “translesional pressure gradient” in a human coronary artery.

**Figure 3 jcm-08-00255-f003:**
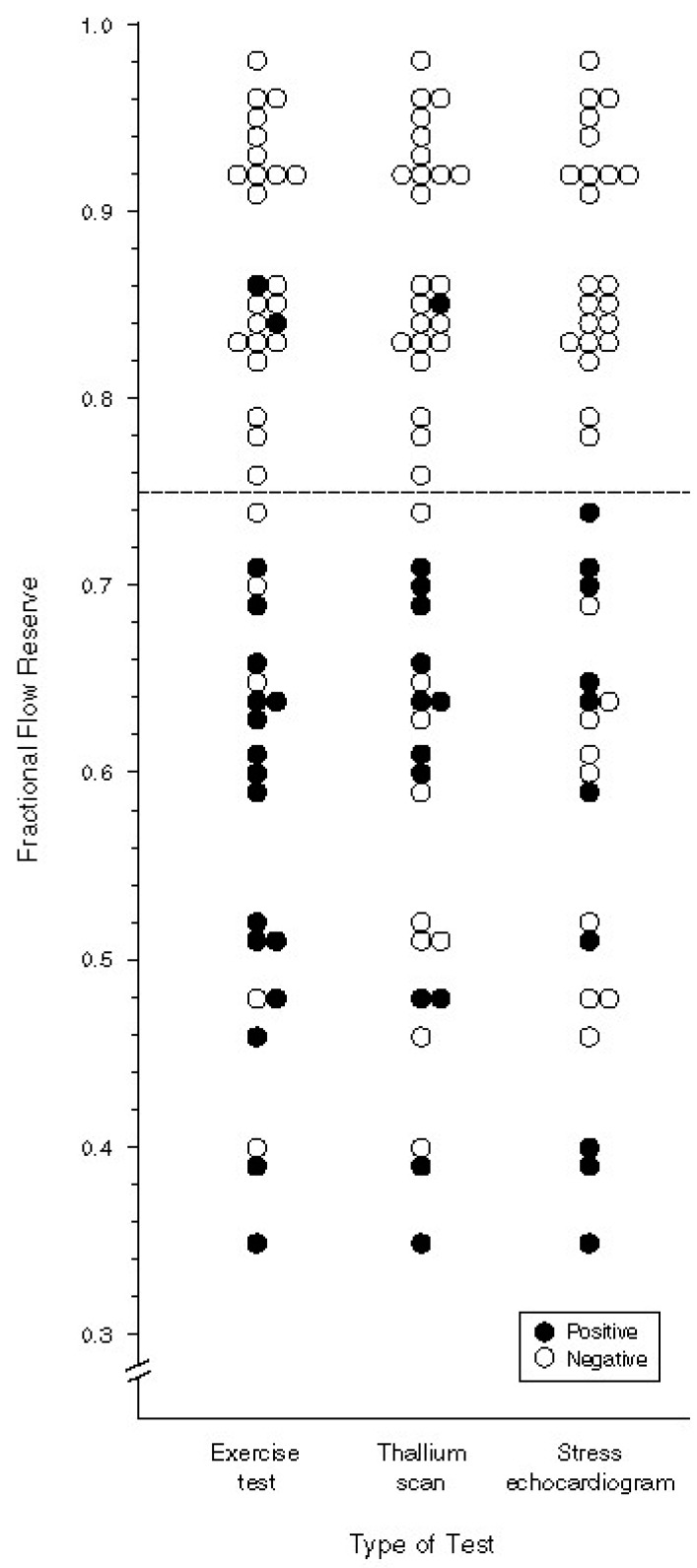
Non-invasive assessment of ischemia compared with FFR. Reprinted with permission from Pijls et al. [22]. In 45 patients with chest pain and moderate (~50%) coronary artery stenosis, exercise stress testing, thallium scintigraphy, and stress echocardiography with dobutamine were performed. These results were compared with FFR measurements. The dashed line indicates the pre-specified FFR threshold for ischemia of 0.75 and each circle represents patients who were found to have ischemia or no ischemia by stress testing. Nearly every patient with ischemia by stress testing was found to have a coronary stenosis with an ischemic FFR value. After revascularization non-invasive tests were repeated and there was no longer evidence of ischemia.

**Figure 4 jcm-08-00255-f004:**
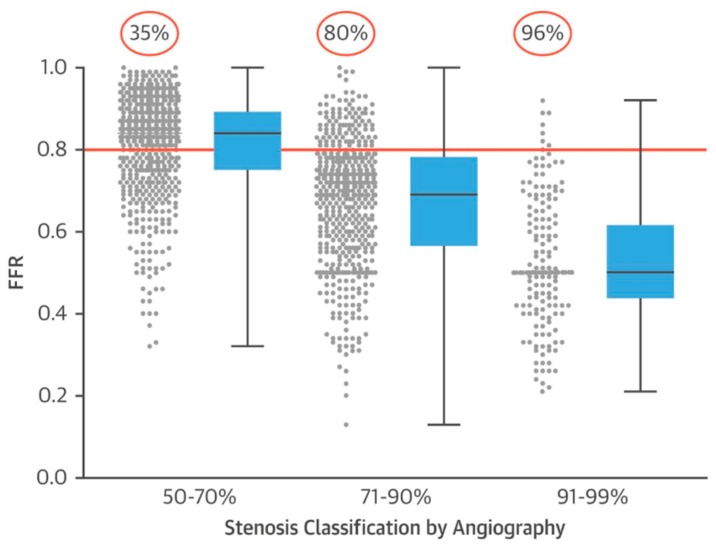
Fractional flow reserve (FFR) values and angiographic severity of lesions in the FAME trial. Reprinted with permission from Jeremias et al., 2017 [14] (which was adapted from Tonino et al 2010) [26]. Angiographic findings of the FFR-guided revascularization arm in the FAME trial [25]. Scatterplot of FFR values shows significant heterogeneity in the physiological significance of angiographic stenoses. Stenoses of 50–70% angiographic severity were found to have an ischemic FFR (0.8, red bar) in 35% of cases. In angiographic stenoses of 71–90%, FFR was in the ischemic zone in 80% of cases. In 91–99% angiographic stenoses FFR was ischemic in 96% of cases.

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
