# Peer review of "Coronary Physiology in the Cardiac Catheterization Laboratory"

_jcm, 2019, doi:10.3390/jcm8020255_

Reviewer 1 Report

This is a narrative review on the diagnostic tools for coronary artery disease available for the cath lab that target to detect insufficiency of coronary physiology. The review presents the theory, technical aspects and relevant clinical trials to a readership of general medical practitioners, and includes physiology indices associated with epicardial artery disease (CFR: coronary flow reserve; FFR: fractional flow reserve; iFR: instantaneous wave-free ratio) as well as indices of microvascular disease in the absence of or in addition to epicardial artery disease (IMR: index microcirculatory resistance; CFVR: coronary flow velocity reserve). The paper is very well written and suitable for a non-specialist audience. While the technological aspects of FFR and iFR determination are well described, the methods to quantify microvasculature resistance by thermodilution and Doppler-wire flow velocity, and how these proximal flow parameters relate to distal microvasculature resistance (page 9) is insufficiently made clear for a non-specialist audience. In this light it may also be useful to discuss the effect of global versus regional perfusion (Gould &Johnson, JACC 2018;72:2642-62). When comparing iFR to FFR (page 8), a reference should be given to the most recent meta-analysis by De Rosa et al (Circ Cardiovasc Interv 2018;11:e04613). The legend to figure 3 is not an accurate reflection of the original paper. FFR and stress testing was performed in patients with chest pain and ~50% stenosis in one of the major coronaries; a subdivision was made between patients that developed signs of myocardial ischemia (positive) or no ischemia (negative) during stress testing. Furthermore, in the legend to figure 4, the fact that Jeremias et al had this figure adapted from Tonino et al (2010) should be acknowledged here. Minor Figure 1: is 1974 in the figure meaningful? Gould 2009 is not included in the reference list. Line 294: Poiseuille References: some but not all titles are in italics. Ref. #28 is not complete.

Author Response

We sincerely appreciate the reviewer's thorough reading and suggested revisions.

Point 1. “While the technological aspects of FFR and iFR determination are well described, the methods to quantify microvasculature resistance by thermodilution and Doppler-wire flow velocity, and how these proximal flow parameters relate to distal microvasculature resistance (page 9) is insufficiently made clear for a non-specialist audience.”

Response 1.

The discussion of coronary flow reserve, index of microcirculatory resistance, and hyperemic microvascular resistance has been expanded, including a discussion of the method for calculating each metric.

Point 2. “In this light it may also be useful to discuss the effect of global versus regional perfusion (Gould &Johnson, JACC 2018;72:2642-62).

Response 2.

This reference is now included (line 273). The authors thoughtfully describe the potential mechanisms for reduced coronary flow reserve (both regional and global) when measured by PET imaging, and this has recently been compared to invasive methods of determining flow reserve/microvascular function. The second reference is cited in line 333.

Point 3. When comparing iFR to FFR (page 8), a reference should be given to the most recent meta-analysis by De Rosa et al (Circ Cardiovasc Interv 2018;11:e04613).”

Response 3.

This reference is now included (line 264).

Point 4. The legend to figure 3 is not an accurate reflection of the original paper. FFR and stress testing was performed in patients with chest pain and ~50% stenosis in one of the major coronaries; a subdivision was made between patients that developed signs of myocardial ischemia (positive) or no ischemia (negative) during stress testing.

Response 4.

The figure legend has been corrected to more accurately reflect the original methodology of the cited reference.

Point 5. Furthermore, in the legend to figure 4, the fact that Jeremias et al had this figure adapted from Tonino et al (2010) should be acknowledged here.

Response 5.

The figure legend has been revised to clarify that the figure was adapted from Tonino et al 2010 and the reference has been cited.

Point 6. Figure 1: is 1974 in the figure meaningful?

Response 6.

The original figure was published in 1974 and the reprinted image carries the label of the year. Unfortunately permissions were obtained for reprinting the image but not modifying it.

Point 7. Gould 2009 is not included in the reference list.

Response 7.

The reference has been added (Line 29).

Point 8. Line 294: Poiseuille References: some but not all titles are in italics.

Response 8.

This section has been revised.

Point 9. Ref. #28 is not complete.

Response 9.

The references have been reviewed for completeness.

Reviewer 2 Report

The present manuscript intends to present a historical perspective and an up-to-date review of the assessment of coronary physiology to the broad audience of general medical practitioners. The manuscript is divided into three main sections which cover the historical basis for the assessment of coronary physiology in the cath lab, the insights into CAD evaluation provided by coronary physiology through the summary of landmark clinical studies (DEFER, FAME, FAME II, ORBITA trials), and the assessment of microvascular disease using well described (IMR) as well as currently evaluated (HMR) techniques. The paper is very well written and should provide the targeted public with a valuable account of the field.

The only comment that this reviewer may have regards the structure of the last two paragraphs of the third section (« Assessment of microvascular disease and endothelial dysfunction »). The authors chose to describe the assessment of microvascular resistance by differentiating thermodilution (CFR and IMR) from Doppler techniques (CFVR and HMR). It seems to this reviewer that describing first the techniques assessing both epicardial and microvascular disease (CFR and CFVR) and then those being developed for the more exclusive assessment of the microcirculation (IMR and HMR) could prove more didactic.

Author Response

We sincerely appreciate the reviewer's thorough reading and suggested revisions.

Point 1. The authors chose to describe the assessment of microvascular resistance by differentiating thermodilution (CFR and IMR) from Doppler techniques (CFVR and HMR). It seems to this reviewer that describing first the techniques assessing both epicardial and microvascular disease (CFR and CFVR) and then those being developed for the more exclusive assessment of the microcirculation (IMR and HMR) could prove more didactic.

Response 1.

The discussion of thermodilution and Doppler derived flow reserve have been expanded with a more robust discussion of coronary flow reserve, index of microcirculatory resistance, and hyperemic microvascular resistance.

Round  2

Reviewer 1 Report

With the revisions made, the manuscript has become much better targeted for a general non-specialist readership. A few minor issues remain:

line 324: Poiseuille is still misspelled;

References: throughout, italics are not used consistenly for paper titles are still in italics

Ref. #28 is still incomplete 

Author Response

We sincerely appreciate the review by the reviewers and academic editor.

Point 1. Please extend the section regarding iFR with the following items points 1) importance of adenosine in the cath-lab to prevent  microcirculation dysfunction and outcomes in particular in patients with acute coronary syndrome;

Response 1: This has been added to the text.

Point 2. 2) accordingly, the concerns on avoid adenosine for the physiological assessment of the stenosis in patients with recent acute coronary syndrome because high risk of microcirculation dysfunction;

Response 2: This has been addressed in the text with a discussion regarding the use of FFR in non-culprit in the setting of acute coronary syndromes as well as the pitfall of using FFR to defer intervention to a culprit stenosis.

Point 3: 3) and, finally, please discuss studies showing the non-inferiority of iFR comparing to FFR in patients with ACS.

Response 3: This is addressed in the discussion of iFR SWEDEHEART from lines 253 to 257.

Point 4: Please include in this section the following articles:
doi: 10.1016/j.ijcard.2014.03.210
doi: 0.1016/j.ijcard.2015.11.086

Response 4: These references have been cited in the text.